# A Cross-Sectional Comparison of Arterial Stiffness and Cognitive Performances in Physically Active Late Pre- and Early Post-Menopausal Females

**DOI:** 10.3390/brainsci12070901

**Published:** 2022-07-09

**Authors:** Amélie Debray, Louis Bherer, Christine Gagnon, Laurent Bosquet, Eva Hay, Audrey-Ann Bartlett, Daniel Gagnon, Carina Enea

**Affiliations:** 1Montreal Heart Institute, 5055 Rue Saint-Zotique E, Montreal, QC H1T 1N6, Canada; amelie.debray@umontreal.ca (A.D.); louis.bherer@umontreal.ca (L.B.); christine.gagnon@icm-mhi.org (C.G.); audrey-ann.bartlett@umontreal.ca (A.-A.B.); daniel.gagnon.3@umontreal.ca (D.G.); 2School of Kinesiology and Exercise Science, Université de Montréal, 2100, boulevard Édouard-Montpetit, Montreal, QC H3T 1J4, Canada; 3Laboratory MOVE (UR20296), Faculté des Sciences du Sport, Université de Poitiers, Batiment C6, 8 allée Jean Monnet, TSA 31113, CEDEX 9, 86073 Poitiers, France; laurent.bosquet@univ-poitiers.fr (L.B.); hayeva86@gmail.com (E.H.); 4Centre de Recherche de l’Institut Universitaire de Gériatrie de Montréal, 4565 Chemin Queen Mary, Montreal, QC H3W 1W5, Canada; 5Department of Medicine, Université de Montréal, Montreal, QC H3T 1J4, Canada

**Keywords:** executive functions, working memory, vascular health

## Abstract

Menopause accelerates increases in arterial stiffness and decreases cognitive performances. The objective of this study was to compare cognitive performances in physically active pre- and post-menopausal females and their relationship with arterial stiffness. We performed a cross-sectional comparison of blood pressure, carotid–femoral pulse wave velocity (cf-PWV) and cognitive performances between physically active late pre- and early post-menopausal females. Systolic (post-menopause—pre-menopause: +6 mmHg [95% CI −1; +13], *p* = 0.27; ŋ^2^ = 0.04) and diastolic (+6 mmHg [95% CI +2; +11], *p* = 0.06; ŋ^2^ = 0.12) blood pressures, and cf-PWV (+0.29 m/s [95% CI −1.03; 1.62], *p* = 0.48; ŋ^2^ = 0.02) did not differ between groups. Post-menopausal females performed as well as pre-menopausal females on tests evaluating executive functions, episodic memory and processing speed. Group differences were observed on the computerized working memory task. Post-menopausal females had lower accuracy (*p* = 0.02; ŋ^2^ = 0.25) but similar reaction time (*p* = 0.70; ŋ^2^ < 0.01). Moreover, this performance was inversely associated with the severity of menopausal symptoms (r = −0.38; *p* = 0.05). These results suggest that arterial stiffness and performance on tests assessing episodic memory and processing speed and executive functions assessing inhibition and switching abilities did not differ between physically active pre- and post-menopausal females. However, post-menopausal females had lower performance on a challenging condition of a working memory task, and this difference in working memory between groups cannot be explained by increased arterial stiffness.

## 1. Introduction

Aging is associated with alterations in structure and function of central and peripheral arteries, which can influence cerebral hemodynamics [1,2] and lead to cognitive impairments [3]. Indeed, central arterial stiffness is associated with reduced cerebral blood flow and consequently hypoperfusion of the brain [4] and is considered as an independent predictor of cognitive decline and cerebrovascular diseases [5,6]. In middle-aged females, several data suggest that menopause could be a critical period in the natural progression of cognitive decline and potentially in future vascular and/or degenerative dementias such as Alzheimer’s disease [7]. First, the drop of estrogen levels during menopause has been hypothesized to contribute to increased blood pressure and arterial stiffness [8], which exposes post-menopausal females to a higher risk of cardiovascular diseases and cognitive decline than their male counterparts [9,10]. Furthermore, menopause is also associated with frequent and persistent symptoms such as vasomotor menopausal symptoms (VMS) that also lead to increased risks of later cardiovascular events [11] and poorer cognitive performance [12]. More precisely, studies have shown that executive functions, heavily supported by the prefrontal cortex, are particularly affected during the menopausal transition [13,14]. However, the specific role of menopause in cognitive decline remains poorly understood. In pre-menopausal females, estrogen appears to contribute to maintaining cognitive function, as shown by a positive correlation between estrogen levels and performance on cognitive tests [15]. Thus, it is possible that estrogen deprivation following menopause could have a negative impact on executive functions by worsening central arterial stiffness [16]. Nonetheless, the mechanisms implicated in cognitive decline remain to be investigated.

Physical activity is associated with both cardiovascular and cognitive health. Regular exercise has been shown to improve vascular function of both conductance and resistance arteries [17] but the extent to which this translates to the cerebral circulation, influences neuronal function or impacts cognition is unclear. Recently, two systematic review and meta-analyses showed that aerobic exercises only do not lead to a clinical improvement in blood pressure in pre- and post-menopausal females. However, when this modality of exercise is combined with resistance exercises, a greater improvement is observed within this population [18,19]. In the same way, it has been shown that one year of aerobic exercise reduces carotid arterial stiffness and increases cerebral blood flow in individuals with amnestic mild cognitive impairment [4]. Therefore, physical activity should be an effective strategy to counteract the deleterious effects of menopause on the brain, especially regarding potential benefits on arterial stiffness (vascular hypothesis), but also by improving brain plasticity (neurogenesis hypothesis) [20]. Unfortunately, a majority of preclinical studies examining the effects of physical activity on vascular components have primarily been performed in males, and the beneficial effects of exercise on central arterial stiffness remain uncertain in post-menopausal females [21]. Moreover, a majority of studies so far have enrolled late post-menopausal females or have not always assessed the participants’ level of physical activity or physical fitness. Therefore, the relationship between blood pressure, arterial stiffness and cognitive performances in healthy and physically active post-menopausal females needs to be further investigated.

The objective of this study was to compare cognitive performances in physically active late pre- and early post-menopausal females and the relationship between these performances and arterial stiffness. The hypothesis was that physically active early post-menopausal females would exhibit similar blood pressure, arterial stiffness and cognitive performances relative to physically active late pre-menopausal females. As an exploratory analysis, the relationship between cognitive performances and menopausal symptoms was also evaluated.

## 2. Materials and Methods

### 2.1. Participants

The data presented herein are secondary outcomes of a previously published study that compared blood pressure and vascular health in physically active late pre- and early post-menopausal females [22]. In the present study, 16 pre- (48 ± 2 years old) and 14 post- (53 ± 2 years old) menopausal females were included. A total of 20 the participants (11 pre and 9 post) were recruited at the Montreal Heart Institute (MHI, Canada), and 10 participants (5 pre and 5 post) were recruited at the Laboratory of Mobility, Aging and Exercise of Poitiers (MOVE, France). Participants were included based on (1) Absence of cognitive impairment (total score for global cognition ≥26/30 on the Montreal Cognitive Assessment (MoCA); and (2) being physically active (≥150 min/week of moderate intensity physical activity or ≥75 min/week of vigorous intensity physical activity). Participants were free of any disease or risk factors for cardiovascular disease, non-smokers, non-obese, and not taking any medication that could alter cardiovascular function.

### 2.2. Study Design

Participants completed one screening and two laboratory visits in a period of 15 days. The first laboratory visit was performed 7 days after the preliminary visit, and both laboratory visits were conducted in a period of 7 to 10 days. During the preliminary visit, medical history and physical examination including measurement of height (wall gauge), weight, body mass index (calculated with the formula: weight (in kg)/height^2^ (in m)) and body composition (bioimpedance, model BC418; Tanita, Arlington, IL, USA) were completed. In addition, a resting ECG and blood pressure measurements were taken. At the end of the preliminary visit, physical activity level was quantified by accelerometery (Actigraph wGT3X- BT, Pensacola, FL, USA) during 7 consecutive days. Participants were advised to avoid strenuous exercise at least 24 h prior to both laboratory visits. For the first laboratory visit, during which arterial stiffness was assessed, participants were also required to arrive fasted (12 h) and to avoid caffeine and alcohol consumption 12 h prior. Pre-menopausal females performed this visit during the first 7 days following the onset of their menses. Participants were instrumented and rested in the supine position for 10 min in a quiet, thermoneutral (~21 °C) laboratory, following which a resting 12-lead ECG and blood pressure measurement were performed. Then, arterial stiffness was assessed by carotid–femoral pulse wave velocity (cf-PWV). During the second laboratory visit, all participants underwent a complete validated neuropsychological test battery to assess cognitive performance. The same day and following cognitive testing, participants performed a maximal cardiopulmonary exercise testing with continuous measurement of gas exchange.

### 2.3. Blood Samples

All blood samples were obtained during the first laboratory visit at arrival in the morning and after a 12 h fast. All blood samples were analyzed by clinical biochemistry laboratories (MHI: Hematology and Biochemistry Laboratory, MOVE: BIO 86 Medical Analysis and Biology Laboratory) for serum glucose, glycated hemoglobin, low-density lipoprotein (LDL), total cholesterol, high-density lipoprotein (HDL), triglycerides, 17ß-estradiol, luteinizing hormone (LH) and follicle stimulating hormone (FSH).

### 2.4. Vascular Assessment

Central arterial stiffness was assessed according to expert guidelines [23]. Pressure waveforms were obtained from the common carotid and femoral arteries with a pencil tip tonometer (MHI: SPT-301 and pressure control units-2000, Millar Instruments, Houston, Texas, USA) or by the Sphygmocor device (MOVE: SphygmoCor v8.0, AtCor Medical, Sydney, NSW, Australia). Transit time between the foot of the carotid and femoral waveforms was determined using a continuously recorded ECG signal. Pressure waveforms were recorded for a minimum of 10 consecutive cardiac cycles. Distance traveled by the pulse wave was also measured in triplicate as the direct distance between the two measurement sites with a correction factor of 0.8 [23]. The cf-PWV was calculated as distance traveled (m) divided by transit time (s).

Systolic and diastolic blood pressures were taken by automated auscultation of the brachial artery (MHI: Tango M2, SunTech Medical, Morrisville, NC, USA; MOVE: Omron M3, Healthcare, Hoofddorp, Haarlemmermeer, The Netherlands). The systolic and diastolic mean value was calculated as the average of three consecutive measurements.

### 2.5. Neuropsychological Assessment

The Montréal Cognitive Assessment (MoCA), a 30-point screening tool, was used to assess global cognition and to confirm inclusion in the study. MoCA subscores include: short-term memory, visuospatial abilities, multiple aspects of executive function, attention, concentration, working memory, language, fluency task and orientation [24]. The cut-off was ≥26/30.

The N-Back task [25] was used to assess updating and working memory. This tablet-based task was performed on an iPad and was administered via the Neuropeak web platform. The stimuli consisted of digits (1 to 9) that were displayed on the center of the screen, one at a time, at a rate of 1.5 s per item. Participants were asked to indicate whether each item matched the one presented in the N position previously (1-, 2- or 3-Back). Each round comprised blocks grouped by N-Back level and performed in the following order: 1-back (4 blocks × 16 trials), 2-back (4 blocks × 16 trials) and 3-Back (2 blocks × 16 trials). Participants reached the 3-Back level only if their accuracy was equal to or above 75% at the 2-Back level. Each level was preceded by a practice that consisted of 3 × 5 trials. If they were below this percentage, they finished the test at the end of the 2-Back level. Participants were instructed to respond as fast as possible without making mistakes. “Match” and “Mismatch” buttons were permanently displayed on the right side of the screen, and participants were required to answer with their right thumb. Measures selected from the N-Back test were accuracy and reaction time to complete each N-Back level. For the N-Back test, two costs were calculated [26]: the 2-Back cost and the 3-Back cost. The 2-Back (1) cost and the 3-Back (2) cost were computed, respectively, as:(2-Back reaction time − 1-Back reaction time)/1-Back reaction time(1)
(3-Back − 2-Back reaction time)/2-Back reaction time(2)

The Digit Span Test (DSST, WAIS-III) was used to assess short-term and working memory by asking participants to repeat series of digits in direct order (forward, condition 1) or backward order (condition 2). The measures collected were the longest number of digits correctly repeated in each condition [27,28].

The Digit Symbol Substitution Test (DSST, WAIS-III) was used to assess processing speed. Participants had to associate specific symbols corresponding to numbers by referring to a response key. They were asked to work as fast as possible. The score provided was the maximum symbols drawn in 120 s [27,28].

The Rey Auditory–Verbal Learning Test (RAVLT) was used to assess verbal learning and memory. The examiner read a list of 15 words and participants were asked to repeat the maximum number of words afterwards. Five successive trials were completed to assess learning capacity. A second 15-word list was then presented for one trial to create interference. Then, participants were asked to recall a maximum number of words from the first list. After a 30-min delay, participants are asked to recall the words from the first list (long-term retention). Measures selected from the RAVLT were the total number of words (out of 75), the number of words (out of 15) retrieved after interference and the number of words recalled after the delay (30 min) [29].

The computerized Stroop task is based on the Modified Stroop Color Test [30] and assesses executive functions including 3 conditions: naming, inhibition and switching. All trials began with a fixation cross for 1.5 s, and all visual stimuli appeared in the center of the computer screen for 2.5 s. Each test consisted of 20 practice trials followed by 60 trials. In the colour-naming condition, participants were asked to read the colour of words that appeared in the center of the computer screen (blue, green, yellow or red) and to provide their responses by pressing the right key on the AZERTY keyboard (“e” for red; “r” for green; “o” for blue; “p” for yellow). Participants were instructed to respond as fast as possible without making mistakes. The inhibition condition required participants to refrain from reading the words to name the incongruent ink colour in which the word was printed (e.g., RED printed in blue ink); 3) in the switching condition. Each test started with the fixation of a symbol (a cross “+” or a square “□”) for 1.5 s, which indicated the task to perform: name the incongruent ink colours (cross) or read (square) the words. This latter condition assessed both inhibition and cognitive flexibility. Measures selected from the Stroop test were mean accuracy and reaction time for each condition. For the Stroop test, two attentional costs were calculated [26]. The inhibition cost (3) and the switching cost (4) were computed as:(Inhibition reaction time − Naming reaction time)/Naming reaction time(3)
(Switching reaction time − Inhibition reaction time)/Inhibition reaction time(4)

The Trail Making Test (TMT, Part A and B) was used to assess speed processing and cognitive flexibility and consisted of circles (including numbers from 1 to 25) distributed over a sheet of paper. In Part A, the participants had to draw lines to connect the numbers in ascending order. In Part B, the circles included both numbers (1–13) and letters (A–L), and the participants had to draw lines to connect the circles in an ascending order, and to alternate between the numbers and letters (i.e., 1-A-2-B-3-C, etc.). The participants were instructed to connect the circles as quickly as possible, without lifting the pen from the paper. The score provided was the time the participants had to connect the “trail”. For the TMT Test, the cost of the switching task (5) was calculated as:(Time TMT B − Time TMT A)/Time TMT A(5)

### 2.6. Maximal Cardiopulmonary Exercise

The maximal cardiopulmonary exercise test was detailed in a previous study by our lab [22]. In brief, maximal cardiopulmonary exercise testing was performed on a treadmill according to the modified Balke protocol [31] to determine maximum oxygen consumption. Continuous 12-lead ECG monitoring was performed at rest and during exercise. Expired gases were continuously measured and analyzed with calibrated gas analyzers (MHI: Cosmed Quark, Albano Laziale, Roma, Italy; MOVE: Metalyzer 3B, Cortex Biophysik GmbH, Leipzig, Saxony, Germany). The analyzers were calibrated using breath-by-breath basis and expressed with a 10 s time averaging for analysis. Standard gases contained 16.0% oxygen and 4.0% carbon dioxide.

### 2.7. Menopausal Symptoms

Menopausal symptoms were assessed using the menopausal rating scale (MRS) [32,33]. This questionnaire regroups 11 items, including hot flashes, heart discomfort, sleep disturbance, depressive mood, irritability, anxiety, physical/mental fatigue, sexual problems, bladder discomfort, vaginal dryness and muscle discomfort. For each item, a scale (0–4 points) was used to evaluate the severity of the complaint (0: none; 1: mild; 2: moderate; 3: severe and 4: very severe). The total score was calculated as the sum of the scores on each item. The severity of menopausal symptoms is divided into 4 levels based on the MRS scores: 0–4 points = asymptomatic, 5–8 points = mild, 9–15 points = moderate and ≥16 points = severe [32,33]. 

### 2.8. Statistics

Data were analyzed using SPSS software (SPSS Statistics, IMB, v27, Armonk, NY, USA), and figures were created using Prism software (Prism, v8, GraphPad, San Diego, CA, USA). All values are reported as mean ± SD except for between-group differences (post—pre menopause) that are presented as means with 95% confidence intervals. A Shapiro–Wilk test was performed to assess the distribution of normality and homoscedasticity by a modified Levene test. Participants’ characterization was obtained through independent-samples *t*-tests (pre-menopausal vs. post-menopausal) comparing both groups’ age, body mass index, years of education and biochemical blood parameters. An analysis of covariance (ANCOVA) with age as a covariate was used to compare mean values of cognitive performance (MoCA, DSST, Digit Span, N-Back, Stroop TMT A and B results), and blood pressure and arterial stiffness values between pre- and post-menopausal females. A repeated-measures ANCOVA was used to compare the 5 RAVLT learning trials. The significance level for all analyses was set at *p* < 0.05. The effect size for analyses of covariance was assessed with partial eta squared (ŋ^2^) and classified as small (ŋ^2^ = 0.01), medium (ŋ^2^ = 0.06) or large (ŋ^2^ > 0.14). The effect size for the independent *t*-test was assessed with Cohen’s d (d) and classified as very small (d < 0.2), small (0.2 < d < 0.5), medium (0.5 < d < 0.8) or large (d ≥ 0.8). Spearman correlations were performed to evaluate the association between systolic blood pressure and cognitive performances and between cf-PWV and cognitive performances. Moderation analyses were used to further understand the association between cf-PWV and cognitive performance through the use of the Hayes macro-application “PROCESS” for SPSS. Specifically, we used a parallel moderation model (model number 1 in Hayes macro).

## 3. Results

### 3.1. Participant Characteristics

Participant characteristics are reported in Table 1. All post-menopausal females experienced a natural menopause and were on average 4 ± 3 years post-menopause. Participants in the post-menopausal group were older (*p* < 0.01) and had greater levels of total cholesterol (*p* = 0.01), low- and high-density lipoproteins (both *p* = 0.01), luteal hormone (*p* < 0.01) and follicle stimulating hormone (*p* < 0.01). Estradiol levels were greater in pre- compared to post-menopausal females (*p* < 0.01). Body mass index, fat and lean body mass, total Cholesterol/HDL ratio, LDL/HDL ratio, triglycerides, moderate to vigorous physical activity levels and maximal oxygen consumption did not differ between groups.

### 3.2. Cardiovascular Assessment

Mean values of systolic, diastolic and pulse pressures and cf-PWV values are presented in Figure 1. The values of systolic (post-menopause—pre-menopause: +6 mmHg [95% CI −1; +13], *p* = 0.27; ŋ^2^ = 0.04), diastolic (+6 mmHg [95% CI +2; +11], *p* = 0.06; ŋ^2^ = 0.12) blood pressures and pulse pressure (0 mmHg [95% CI −5; +5], *p* = 0.75; ŋ^2^ < 0.01) did not differ between pre- and post-menopausal females. The cf-PWV did not differ between groups (+0.29 m/s [95% CI −1.03; 1.62], *p* = 0.48; ŋ^2^ = 0.02).

### 3.3. Neuropsychological Assessment

All results are presented in Table 2. Due to technical difficulties, six participants have been excluded from the analyses for the N-Back test, and one did not reach 75% accuracy at levels 1 and 2. Both groups performed similarly on the clinical neuropsychological tests: Digit Span Test forward and backward, the Stroop Test, the Digit Symbol Substitution Test, the Trail Making test A and B and the Rey Auditory Verbal Learning Test. Group differences were observed in the computerized working memory task (N-Back). Post hoc comparisons (Bonferroni) showed that post-menopausal females had lower accuracy (*p* = 0.02; ŋ^2^ = 0.26) but similar reaction time (*p* = 0.81; ŋ^2^ < 0.01) for the 3-Back condition compared with pre-menopausal females (Table 2 and Figure 2).

### 3.4. Correlation Analyses

Cf-PWV was not associated with cognitive performance for 3-Back accuracy (r = −0.03; *p* = 0.89) when combining participants from both groups. When the analysis was performed separately for each group, this relationship was not significant in pre-menopausal (r = −0.11, *p* = 0.76) nor in post-menopausal (r = 0.09, *p* = 0.80) females (Figure 3). Cf-PWV was also positively associated with reaction time for naming and inhibition (both, r = 0.41; *p* = 0.02) and switching (r = 0.50; *p* = 0.01) conditions in the Stroop task when pooling participants from both groups. SBP was negatively associated with accuracy in the inhibition condition of the Stroop task (r = −0.42; *p* = 0.02) and positively associated with reaction time in inhibition condition (r = 0.3; *p* = 0.04) and was negatively associated with immediate and delayed recalls (both, r = −0.39; *p* = 0.04). When the analyses were performed separately for each group, these relationships were not significant in either group.

We found a negative correlation (r = −0.38; *p* = 0.05) between accuracy on the 3-Back task and total menopausal symptoms (Figure 4) in both groups combined such that greater menopausal symptoms were associated with lower accuracy. When focusing on vasomotor symptoms only, we did not find any correlation (Figure 5) in all participants (r = −0.09, *p* = 0.69) or in each group separately. Finally, we did not find any significant correlation between cognitive performances and cardiorespiratory fitness nor with moderate to vigorous physical activity levels.

### 3.5. Moderation Analysis

To further explore the relationships between age, cf-PWV, cognitive performance and estradiol concentrations, we performed moderation analyses (data not presented). However, estradiol concentrations did not appear to moderate the relationship between cf-PWV and cognitive performances or between blood pressure and cognitive performances.

## 4. Discussion

The main objective of this study was to compare cognitive performances between physically active late pre- and early post-menopausal females and the relationship between these performances and arterial stiffness. We hypothesized that physically active early post-menopausal females would exhibit similar blood pressure, arterial stiffness and cognitive performances relative to physically active late pre-menopausal females. The main results of our study highlighted that blood pressure and arterial stiffness do not differ between physically active post-menopausal females compared with late pre-menopausal females. Both groups also showed similar results on neuropsychological tests assessing cognitive flexibility, inhibition, episodic memory and processing speed. However, post-menopausal females had poorer performances on a challenging computerized working memory task than pre-menopausal females. This observation suggests that using more sensitive cognitive tests might allow the detection of subtle cognitive changes in the menopausal transition of physically active late pre- and early post-menopausal females. However, these cognitive differences are not related to blood pressure or arterial stiffness. 

### 4.1. Menopause, Physical Activity and Arterial Stiffness

In sedentary females, menopause is usually associated with an increase in blood pressure and arterial stiffness [8,34] that contribute sometimes to decreased cognitive functions [3,5]. To date, it remains unclear whether a physically active lifestyle and high fitness make it possible to limit the deleterious effects of menopause usually observed in sedentary females. Our results show that carotid–femoral pulse wave velocity did not differ between physically active late pre-menopausal females compared with early post-menopausal females. Previous results from interventional studies [35,36,37] suggested a protective effect of physical activity on arterial stiffness in late post-menopausal females. In cross-sectional studies, arterial stiffness has been shown not to differ between healthy physically active (>6 h per week or training for 13 ± 1 years) pre- and post-menopausal females [34]. Furthermore, other results have also highlighted a negative correlation between arterial stiffness assessed by cf-PWV and physical activity levels in healthy post-menopausal females with normal weight [38]. Nonetheless, these studies have several limits. The first included post-menopausal females ~30 years older than pre-menopausal females and did not mention how physical activity was quantified [34], while the second study quantified physical activity by questionnaire. To date, this relationship has not been tested in early post-menopausal females. Our results extend these findings to early post-menopausal females with a small age difference (5 years) and with an objective quantification of physical activity. In addition, changes in lipid profiles observed during menopausal transition have been shown to contribute to increase atherosclerosis development and cardiovascular disease [39]. In a recent observational study including adults free of cardiovascular disease, higher total cholesterol/HDL-c and LDL-c/HDL-c ratios were observed in adults having higher intima–media thickness, an important predictor for atherosclerosis and closely related to arterial stiffness, compared with those that had a lower intima–media thickness [40]. In our study, total cholesterol/HDL and LDL/HDL ratios did not differ between our groups of pre-and post-menopausal females (respectively, *p* = 0.40 and *p* = 0.26), and both of these markers are considered better predictors for atherosclerosis [40]. This could be a reason why we did not observe differences in vascular structure/function between pre- and post-menopausal females.

### 4.2. Menopause, Arterial Stiffness and Executive Function

Arterial stiffness, assessed by pulse wave velocity, has been associated with changes in cerebrovascular reactivity and cognitive decline with aging [41]. Executive functions are particularly affected by vascular aging. In adults, data from a recent meta-analysis showed a negative association between arterial stiffness and cognition, specifically executive function, memory and global cognition [5]. However, several limits can be highlighted from this meta-analysis. The authors did not separately analyze according to participants’ sex, and the majority of studies reported non-healthy adults (with impaired glucose tolerance, kidney disease, memory loss, hypertension or stroke) [5]. When sex-related differences are analyzed, this relationship is found in older males, with higher pulse wave velocity associated with lower global cognition and memory, but not in older females [41]. Thus, the specific effects of arterial stiffness on cognition remain unclear in healthy post-menopausal females. 

Our results show that physically active post-menopausal females perform as well as their pre-menopausal counterparts on tests evaluating executive functions (inhibition and switching abilities). However, post-menopausal females performed worse on the more challenging condition of a computerized working memory task (3-Back). In the literature, several studies examined the effects of estrogen on working memory performances in post-menopausal females and found advantages associated with hormonal replacement therapy compared to placebo [42,43]. Post-menopausal females treated with hormonal replacement therapy performed significantly better on tests assessing working memory than untreated post-menopausal females. Our results are difficult to compare with these previous results. Indeed, in these studies, post-menopausal females were at a late stage of menopause (~9 years). Moreover, these studies did not consider the participants’ level of physical activity. Our results extend these previous findings to younger, physically active post-menopausal females. In the present study, cardiovascular parameters did not differ between late pre- and early post-menopausal females, and we did not find any relationship between these parameters (including systolic and diastolic blood pressures, pulse pressure and cf-PWV) and working memory performances. Thus, the difference we observed is not related to vascular parameters and could be explained by hypotheses other than vascular health. As suggested in other studies, the loss of the neuroprotective and anti-aging properties of estrogen in post-menopausal females could explain this difference. Indeed, experimental and epidemiological studies suggest that menopause leads to the cessation of exposure to female sex hormones and could thus impact late-life cognitive function. However, evidence remains controversial [44], and the effects of estrogen on cognitive functions still requires further investigation [45].

### 4.3. Menopause and Episodic Memory

Menopause is associated with increased forgetfulness and “brain fog” earlier in the aging process, and post-menopausal females have an increased risk for memory disorders later in life [46]. A recent study showed that deficits in verbal episodic memory learning before age 50 could predict further cognitive impairment in adults over 65 years of age [47]. Episodic memory may also reflect disruptions of attentionally mediated memory processes (encoding and retrieval) as opposed to true retentive memory loss. By evaluating episodic memory, we wanted to explore if subjective memory loss commonly reported in peri-menopause is also observed in physically active females early in menopause. Our results show that physically active post-menopausal females perform as well as active pre-menopausal females on tests assessing episodic memory. In post-menopausal females, estrogen levels are not clearly related to episodic memory, and females taking hormonal replacement therapy do not perform especially better on tests evaluating episodic memory [48]. As observed in a randomized controlled study involving 180 healthy and naturally post-menopausal females, no significant between-group differences on episodic memory were observed between females under hormonal replacement therapy compared with those free of hormonal replacement therapy [49].

### 4.4. Menopausal Symptoms and Cognitive Performance

The abrupt changes in estrogen levels during menopause are associated with the occurrence of menopausal symptoms and cognitive decline. Indeed, these fluctuations have a significant impact on the whole body, including the central nervous system, and can be responsible for modifications in behavior, cognition and mood [50]. Vasomotor symptoms, such as hot flashes and night sweats, are the most common symptoms and affect 50% to 80% of post-menopausal females [51,52,53]. These symptoms, mediated by vascular health, are also associated with increased cardiovascular risk [54] and cognitive decline [46] in post-menopausal females. When considering the entire sample, our results show a negative correlation between working memory performance in the 3-Back condition and the severity of total menopausal symptoms. When the analysis was performed separately for pre- and post-menopausal females, this relationship was not significant (Figure 3). Furthermore, when the analysis was performed for vasomotor symptoms only, we did not find any relationship in the combined group nor in each group separately. Few studies have explored the relationship between total menopausal symptoms and cognitive performances and even less so focusing specifically on vasomotor symptoms. In the literature, this relationship remains unclear. Previously, a study showed that subjectively reported hot flashes were related to cognitive decline (increased sleep disturbances, anxiety and depression scores, decreased executive function) in healthy post-menopausal females [55]. Another study showed no relationship between memory symptoms during peri-menopause and retentive memory performance. However, memory complaints were associated with poorer memory encoding and increased depressive symptoms [56]. Furthermore, physical activity has been relatively well demonstrated to reduce menopausal symptoms, particularly vasomotor symptoms [57,58]. However, women’s physical activity levels are rarely reported. Because our results do not appear to be mediated by vascular health, we investigated whether this correlation was mediated by hormonal factors. However, we did not observe a correlation with estrogen concentrations, nor with FSH and LH concentrations. To date, this relationship remains misunderstood. 

### 4.5. Limits and Perspectives

Several limits should be considered when interpreting these results. First, we used a cross-sectional design. It is possible that the lack of difference between groups may not only result from menopausal status. Second, we did not include groups of sedentary pre- and post-menopausal females. In this sense, we cannot directly determine whether being physically active following menopause offsets some of the physiological and cognitive changes that occurred during the menopausal transition. In addition, pre-and post-menopausal females were highly active, and this high level of physical activity is not reflective of the population that usually follows the recommendation physical activity. Third, the data presented herein are secondary outcomes of a previously published study that compare blood pressure and vascular health in physically active late pre- and early post-menopausal females. Indeed, it is possible that this study was not powered enough to detect a potential change in cognitive function. Fourth the small sample size (increase in type II error) and multiple testing bias (increase in type I error) should also be taken into consideration in the limits of this study. Finally, women included are relatively young and less likely to have poorer cognitive function at baseline.

Future studies should seek to understand the effects of physical activity on the underlying mechanisms involved in the manipulation of working memory since these mechanisms, in addition to being more sensitive to aging, could also be more sensitive to the hormonal changes associated with menopause. From the current study design, it cannot be determined whether these differences are indicative of more prominent alterations in cognitive function that may manifest with further aging. Future studies are required to determine whether high levels of physical activity simply delay the onset of cognitive alterations associated with the menopausal transition or whether they can provide a longer-term protective effect. Additional research focusing on the effects of physical activity on vascular health and brain plasticity during and shortly after the menopausal transition will ultimately help to improve primary prevention efforts to minimize cerebrovascular risk in post-menopausal females. 

## 5. Conclusions

With this cross-sectional design, this study showed that arterial stiffness and performance on tests assessing episodic memory, processing speed and executive functions assessing inhibition and switching abilities did not differ between physically active pre- and post-menopausal females. However, post-menopausal females had lower performance on a challenging condition of a working memory task. Furthermore, we found that performance on the working memory task negatively correlated with total menopausal symptoms. Finally, our results also suggest that this difference in working memory between groups cannot be explained by increased arterial stiffness.

## Figures and Tables

**Figure 1 brainsci-12-00901-f001:**
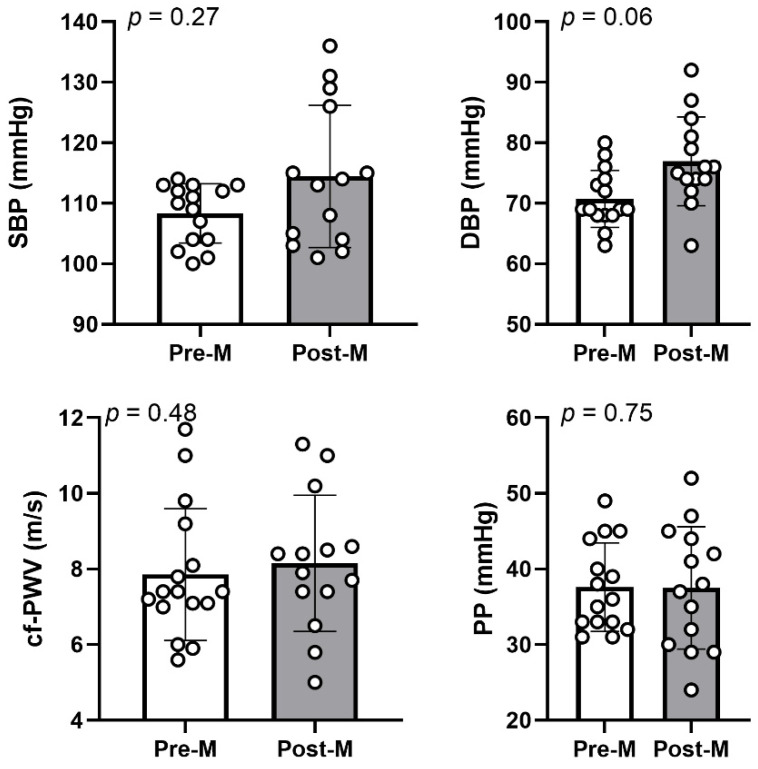
**Blood pressure and central stiffness assessed in physically active pre- (Pre-M, white) and post- (Post-M, grey) menopausal females**: SBP, systolic blood pressure (top left panel); DBP, diastolic blood pressure (top right panel); cf-PWV, carotid-femoral pulse wave velocity (bottom left panel); PP, pulse pressure (bottom right panel). Data are presented as mean ± SD with individual values for 16 pre- and 14 post-menopausal females. *p* value is for an ANCOVA with age as a covariate.

**Figure 2 brainsci-12-00901-f002:**
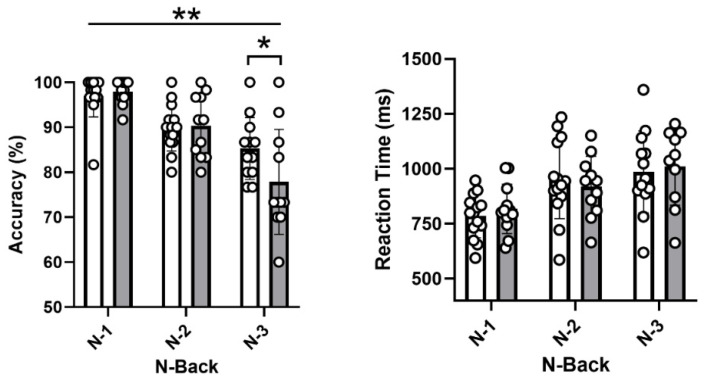
**N-Back accuracy (left panel) and reaction time (right panel) in physically active pre- (white bar) and post- (grey bar) menopausal females**. Data are presented as mean ± SD with individual values for 13 pre- and 10 post-menopausal females. The *p* value is for an ANCOVA with age as covariate. * *p* < 0.05; ** *p* < 0.01.

**Figure 3 brainsci-12-00901-f003:**
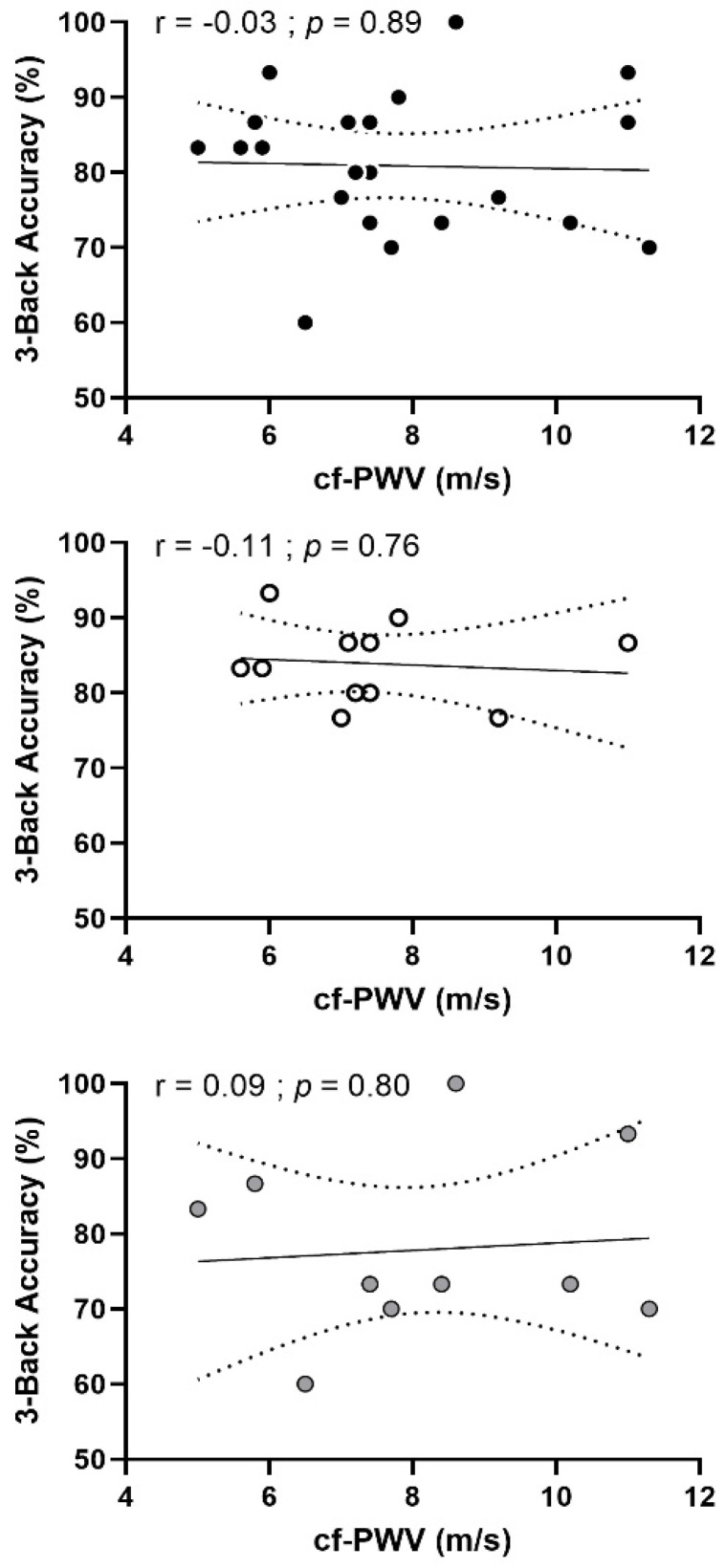
**Relationship between 3-Back accuracy and cf-PWV in pre- and post-menopausal females.** Top panel: pre-menopausal (n = 13) and post-menopausal (n = 10) females combined. Middle panel: pre-menopausal females only. Bottom panel: post-menopausal females only. The r and *p* values are for a Pearson correlation analysis.

**Figure 4 brainsci-12-00901-f004:**
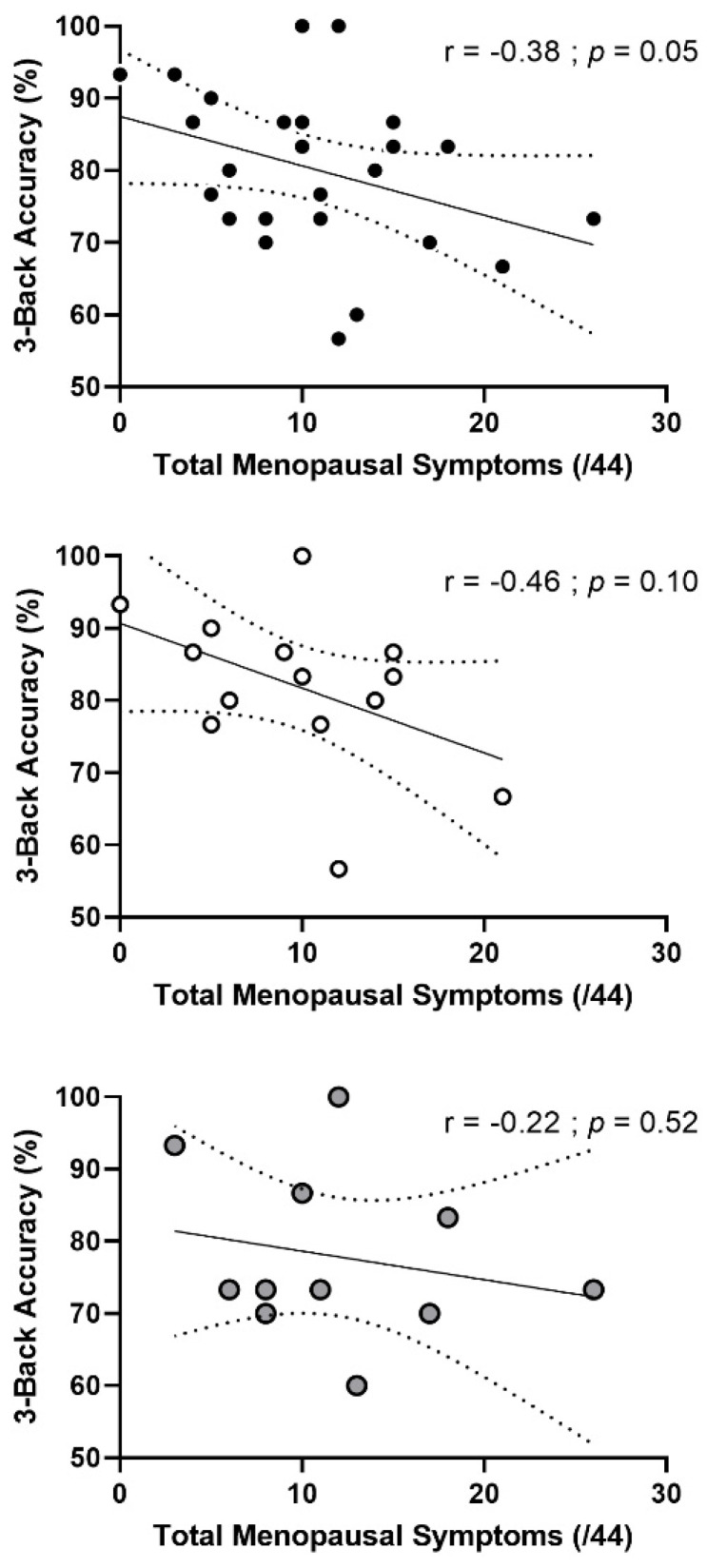
**Relationship between accuracy in 3-Back test and total menopausal symptoms (top panels) in pre- and post-menopausal females**. Top panel: pre-menopausal (n = 13) and post-menopausal (n = 10) females combined. Middle panel: pre-menopausal females only. Bottom panel: post-menopausal females only. The r and *p* values are for a Pearson correlation analysis.

**Figure 5 brainsci-12-00901-f005:**
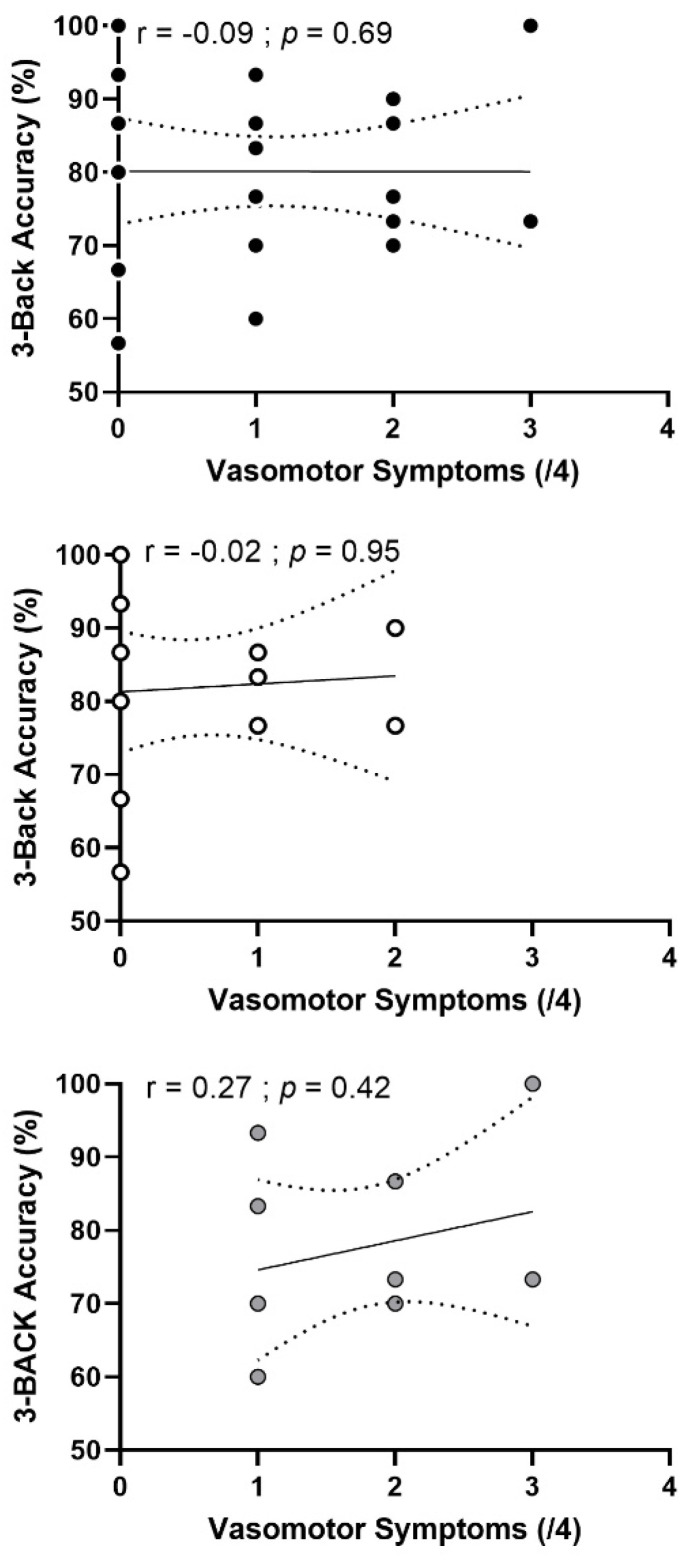
**Relationship between accuracy in 3-Back test and vasomotor symptoms (top panels) in pre- and post-menopausal females.** Top panel: pre-menopausal (n = 13) and post-menopausal (n = 10) females combined. Middle panel: pre-menopausal females only. Bottom panel: post-menopausal females only. The r and *p* values are for a Pearson correlation analysis.

**Table 1 brainsci-12-00901-t001:** Participant characteristics.

	Pre-Menopause (n = 16)	Post-Menopause (n = 14)	*p* Value	Cohen’s d
**Age (years)**	48 ± 2	53 ± 2	<0.01	>0.8
**Education (years)**	19 ± 3	19 ± 3	0.51	<0.2
**BMI (kg/m^2^)**	23.8 ± 3.5	22.2 ± 2.8	0.19	<0.2
**Fat mass (%)**	29.1 ± 5.5	28.4 ± 7.1	0.74	<0.2
**Body mass (kg)**	63.5 ± 9.7	59.9 ± 8.8	0.52	<0.2
**Total cholesterol (mmol/L)**	4.8 ± 0.7	6.2 ± 1.6	<0.01	0.8
**LDL (mmol/L)**	2.5 ± 0.6	3.5 ± 1.3	0.01	1.0
**HDL (mmol/L)**	1.9 ± 0.3	2.3 ± 0.4	0.02	0.9
**Triglycerides (mmol/L)**	0.9 ± 0.3	0.8 ± 0.3	0.64	<0.2
**Total Cholesterol/HDL**	2.6 ± 0.5	2.7 ± 0.5	0.40	0.3
**LDL/HDL**	1.3 ± 0.5	1.5 ± 0.4	0.26	0.4
**Estradiols (pmol/L)**	267.0 ± 179.1	<36.7	<0.01	<0.2
**MVPA (min/week)**	490 ± 214	550 ± 303	0.87	0.2
**VO_2_max (mL/min/kg)**	35.8 ± 6.0	34.0 ± 6.1	0.18	<0.2

Data are presented as mean ± SD: BMI, body mass index; VO_2_max, maximal oxygen consumption; LDL, low-density lipoprotein; HDL, high-density lipoprotein; MVPA, moderate to vigorous physical activity. *p* value is for an independent samples *t*-test.

**Table 2 brainsci-12-00901-t002:** Neuropsychological variables.

Neuropsychological Tests	Pre-Menopause (n = 16)	Post-Menopause (n = 14)	*p* Value	Eta Squared ŋ^2^
**MoCA (/30)**	29 ± 1	28 ± 1	0.26	0.05
**DSST (/133)**	88 ± 14	84 ± 12	0.51	0.02
**TMT A (s)**	27.2 ± 8.5	29.6 ± 8.6	0.06	0.12
**TMT B (s)**	51.5 ± 11.7	56.0 ± 14.6	0.40	0.03
**TMT (B-A)/A**	0.97 ± 0.46	0.98 ± 0.52	0.52	0.02
**EMPAN forward (/8)**	5.5 ± 0.7	5.4 ± 1.2	0.26	0.05
**EMPAN backward (/7)**	4.2 ± 1.3	3.9 ± 1.3	0.07	0.11
**Stroop test—Accuracy (%)**				
**Naming**	99.0 ± 1.5	98.3 ± 2.4	0.87	<0.01
**Inhibition**	99.2 ± 1.0	98.4 ± 1.9	0.72	<0.01
**Switching**	92.0 ± 6.4	87.6 ± 12.2	0.88	<0.01
**Stroop test—Reaction Time (ms)**				
**Naming (N)**	727 ± 113	766 ± 160	0.33	0.04
**Inhibition (I)**	783 ± 72	863 ± 158	0.54	0.01
**Switching (S)**	1092 ± 182	1263 ± 289	0.58	0.01
**Stroop (I-N)/N**	0.10 ± 0.16	0.15 ± 0.21	0.60	0.01
**Stroop (S-I)/I**	0.40 ± 0.22	0.48 ± 0.27	0.95	<0.01
**N-Back test—Accuracy (%)**				
**1-Back**	96.8 ± 4.6	96.1 ± 5.7	0.98	<0.01
**2-Back**	89.9 ± 5.2	90.3 ± 6.7	0.84	<0.01
**3-Back**	83.9 ± 8.0	77.9 ± 11.1	0.02	0.36
**N-Back—Reaction Time (ms)**				
**1-Back**	780 ± 98	800 ± 95	0.80	<0.01
**2-Back**	951 ± 179	918 ± 133	0.46	0.03
**3-Back**	1047 ± 259	1055 ± 214	0.46	<0.01
**N-Back (2-1)/1**	0.23 ± 0.26	0.15 ± 0.10	0.60	0.01
**N-Back (3-2)/2**	0.10 ± 0.13	0.15 ± 0.16	0.95	<0.01
**RAVLT**				
**Rey 1–5 total** **words (/75)**	58 ± 8	55 ± 8	0.61	0.09
**Immediate recall (/15)**	13 ± 2	12 ± 2	0.73	<0.01
**Delayed recall (/15)**	13 ± 1	12 ± 2	0.61	0.01

Data are presented as mean ± SD. The *p* value is for an ANCOVA with age as a covariate. MoCA, Montreal Cognitive Assessment; DSST, Digit Symbol Substitution Test; TMT, Trail Making Test; RAVLT, Rey Auditory Verbal Learning Test.

## Data Availability

Data supporting reported results can be found here: https://doi.org/10.6084/m9.figshare.20264040.v1 (accessed on 3 July 2022).

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
