# Peer review of "A Cross-Sectional Comparison of Arterial Stiffness and Cognitive Performances in Physically Active Late Pre- and Early Post-Menopausal Females"

_brainsci, 2022, doi:10.3390/brainsci12070901_

Round 1

Reviewer 1 Report

I have reviewed the Debray et al. paper entitled “A cross-sectional comparison of arterial stiffness and cognitive performances in physically active late pre-and early post-menopausal females.”. The study is interesting and can merit publication if some issues are approached. Here are my comments:

-Given the very small sample size and the propensity to multiple testing bias, this study is hypothesis-generating and this should be addressed in the Abstract and Conclusions sections. This is of major importance, since applying the Bonferroni method for multiple testing adjustments, the N-Back test performance between groups would have a non-significant difference (about 0.50). This does not invalidate results per se, but the authors should lower the tone of the conclusion suggesting that this is a pilot sub-study to generate hypotheses and further work. 

-Cholesterol variables differ between groups, but not the vascular structure/function parameters. It is necessary to discuss the possible causes of these findings.

-The introduction section can be shortened to address only the most important background elements.

-I would recommend simplifying the sentence about the exploratory hypothesis (lines 90–93).

-A study limitations section is advisable (before Conclusion) to improve the correct interpretation of the results, addressing the very small sample size (i.e., increase in type II error) and multiple testing bias (i.e., increase in type I error).

-Line 589: reference 23 seems incomplete or with errors.

Reviewer 2 Report

Authors

Line 13. Please spell "Faculté" correctly

Abstract.

This section allows the future reader to get an idea of what research has been carried out; how it has been carried out; the subjects who have participated as a study sample; the main results and finally an adequate approximation of the conclusions that the researchers have interpreted once their work has been completed.

Introduction

Lines 40-41; 41-42; 43-44... Please avoid breaking the syllables at the end of each line. This is a spelling mistake and although we know that the template of the journal allows this break, it is still a spelling mistake that affects your prestige as a writer. Please check it throughout the text and change it.

Lines 66-70. Please consider including the following quote in which quality of life and blood pressure are studied in a similar sample of women. Thank you in advance. Jiménez-Gómez, S.; Sánchez Rojas, I. A.; Castro-Jiménez, L.; Rubiano-Espinosa, O.; Carrillo-Ramírez, C.; Garavito-Peña, F.; Barrera-Cobos, N. Quality of life in assistants to a physical activity program in Bogotá, Colombia. RICCAFD 2021, 10, 95-111.

Lines 66-72. The effects that he indicates on the practice of "regular exercise" do not correspond to the conclusions of the articles he cites. Not all types of exercise exert this benefit. For example, these are the conclusions of his quote number 20: Aerobic Training did not lead to a clinically relevant improvement in blood pressure (BP) in this population. In addition, Combined Traininning showed the largest reductions in SBP, DBP and MAP.

Please review these arguments.

In general, the introduction gives a good review of the state of the art and focuses on previous work very close to the object of study of this research. In addition, they consciously focus this work on the under-studied population of pre-post menopausal women.

2. Materials and Methods

Participants

Line 95. Please delete the dot after "Participants."

Please report on the informed consent of the participants, the ethical conduct of the research and provide the code of ethics approval of your university's ethics committee for this experiment.

Study design

Line 108. Please delete the dot after "Study design."

Information on size, measuring instrument and degree of accuracy is missing. The Tanita model BC418 does not measure height.

It is not sufficiently clear whether 7 or more days elapse between pre-test and post-test. It is only reported that they had the accelerometer on for 7 consecutive days, but we do not know how many days before the pre-test and how many days after the 7 days the post-test was taken. This needs to be better explained.

Blood samples 

Line 127. Please delete the dot after "Blood samples."

Please report the amount of blood drawn and the times at which it was drawn.

Vascular assessment

Please delete the bullet after "Vascular assessment." Line 134.

Line 134. "Central arterial stiffness" is not justified to be italicised. Please change it to normal font.

Line 135-136. Please justify why different instruments have been used to measure "pressure waveforms". Include reports of variability between instruments and differences in accuracy.

Neuropsychological assessment

Line 147. Please delete the dot after "Neuropsychological assessment."

Excellent selection and explanation of tests

Maximal cardiopulmonary exercise

Line 199. Please delete the bullet point after "Maximal cardiopulmonary exercise."

Line 204-206. Please justify why different instruments have been used to measure gases. Include reports of variability between instruments and differences in accuracy.

Menopausal symptoms

Line 215. Please remove the full stop after "Menopausal symptoms."

Excellent explanation of this section.

Data analyses

Line 215. Please delete the bullet point after "Data analyses."

Lines 226-235. I feel that these explanations should be included in the previous sections when explaining these tests.

Statistics

Line 236. Please delete the bullet point after "Statistics."

Line 234. At the end of this line you report that BMI has been taken into account, however you have not previously reported how you have measured, and with instrument, the subjects participating in the experiment. Please this should be included.

3. Results

3.1 Participant characteristics

Line 259. Please delete the dot after "Participant characteristics." Line 270.

Line 270. It is unclear whether the characteristics that have been stated in Table 1 are at the beginning of the experiment, at the end or is an average of the two points in time. This should be clarified.

Please include the effect size in the tables for all results.

3.2 Cardiovascular assessment

Line 275. Please delete the dot after "Cardiovascular assessment."

Line 277-290. It is not clear whether the variables that have been displayed in Figure 1 are at the beginning of the experiment, at the end or an average of the two points in time. This needs to be clarified.

Please include the effect size in the report of the results.

3.3 Neuropsychological assessment

Line 275. Please remove the dot after "Neuropsychological assessment."

Line 294-312. It is not clear whether the variables that have been displayed in Table 2 and Figure 2 are at the beginning of the experiment, at the end or an average of the two points in time. This needs to be clarified.

Please include the effect size in the report of the results.

3.4 Correlation analyses

Line 275. Please remove the dot after "Correlation analyses."

The authors make an excellent effort to present an explanation between variable correlations.

On lines 98-99 the subjects are stated to be 16 pre- and 14 post menopausal females, however, the information on line 337 now states 13 and 10 respectively and also on lines 343-344 and 349-350. This may have altered the results and should be explained.

In this section of the results, variables that are indicated as having been collected, such as the subjective perception of fatigue (line 203: Ratings of perceived exertion (RPE, Borg 203 scale), do not appear.

4. Discussion

The authors have made a good division into parts by dealing with groups of variables which will help the future reader to better locate the findings of this research.

5. Conclusions

The recommendations of international health institutions have recommended volumes of moderate-intense physical activity well below what has been recorded with accelerometry in this research (Table 1) and furthermore in this variable there are no differences between the groups, to which we must add that we do not know the size of the effect size and it cannot be concluded that "working memory differs between physically active late pre- and early post-menopausal females".
